# Learning Oscillatory Brain Dynamics using Nonlinear Kalman Smoothing

**Germán Abrevaya**
University of Buenos Aires
germanabrevaya@gmail.com

**Aleksandr Aravkin**
University of Washington `saravkin@uw.edu`

**Guillermo A. Cecchi, Pablo Polosecki, & Irina Rish**
IBM Thomas J. Watson Research Center
{gcecchi, pipolose, rish}@us.ibm.com

## Abstract

Motivated by the challenging problem of modeling nonlinear dynamics of brain activations in calcium imaging, we propose a novel approach for learning a nonlinear differential equation model: a variable-projection optimization approach for estimating the parameters of the multivariate (coupled) van der Pol oscillator. To the best of our knowledge, we are the first to propose a successful approach to learning such nonlinear dynamical model, and to demonstrate that it can accurately capture nonlinear dynamics of the brain data. Furthermore, in order to further improve the predictive accuracy when forecasting future brain activity, we use the learned analytical van der Pol model to generate large amounts of simulated data for LSTM pretraining, since brain imaging datasets are often limited in size, and since the generic LSTM model may benefit from the oscillator prior imposed via such pretraining. Indeed, the proposed combination of the analytical approach with the general-purpose statistical LSTM model improves the performance of both methods.

## 1 Introduction

Neuroscience is a source of exceedingly interesting and challenging data for analysis and modeling, generated by the underlying multivariate coupled nonlinear dynamical system, a.k.a. the brain. Brain activity clearly exhibits a highly nonlinear, oscillatory and periodic (or nearly periodic) behavior, with sharp phase transitions between different states; thus, modeling such time series can be nontrivial. Also, state-of-art temporal models such as LSTM networks may require large amounts of training data, not always readily available in biological applications, in order to capture complex nonlinear dynamical processes; moreover, such models generally lack interpretability.

In this work, we consider brain-wide calcium imaging larval zebrafish data Ahrens & et al (2013) and propose a novel approach for estimating the parameters of van der Pol oscillator – an analytical model governed by nonlinear ordinary differential equations, which describe calcium dynamics in the brain using both observed voltage-like variables, reflecting voxel activity, and hidden recovery-like variables, reflecting excitability. To the best of our knowledge, we are the first ones to propose an approach to this problem, and demonstrate its successful application on real data. Note that fitting parameters of *linear* state space systems is a well-studied problem. However, inferring parameters from nonlinear ODEs requires more care, and we develop a specialized algorithm that is different from the classic EM approach typically used for linear systems. Namely, we propose a variable-projection optimization approach to estimate the parameters of the multivariate (coupled) van der Pol oscillator, and demonstrate that the proposed model can accurately capture the nonlinear dynamics of the brain data, and provide interpretable results as it learns a coupling matrix indicating interactions across brain areas.

Furthermore, we improve the accuracy of predicting future brain activity using a combination of both analytical van der Pol model and general-purpose statistical model such as LSTM, which outperforms both techniques considered separately. Namely, the van der Pol model learned on relatively small

training dataset is used as a simulator of unlimited pre-training data which effectively impose an oscillatory "prior" on LSTM, i.e. a kind of regularizer which directs LSTM model towards specific oscillatory patterns we assume are present in the data.

## 2 OUR APPROACH

**Whole-Brain Calcium Imaging Data.** In Ahrens & et al (2013), light-sheet microscopy was used to record the neural activity of a whole brain of the larval zebrafish, reported by a genetically-encoded calcium marker, in vivo and at 0.8 Hz sampling rate. From the publicly available data Channel (2013) it is possible to obtain a movie of 500 frames with a 2D collapsed view of 80% of the approximately 40,000–100,000 neurons in the brain, with a resolution of 400 by 250 pixels (approximately 600 by 270 microns). We performed SVD to reduce the dimensionality of the data and we used the top 6 components explaining most of the variance (spatial components corresponding to brain areas; we learn a coupled oscillator on time components).

**Van der Pol Model.** Because neuronal calcium dynamics are largely driven by transmembrane voltage and voltage-dependent calcium channels, we propose to describe the calcium dynamics of a neuron, or small clusters of them, as a 2D differential equation with a voltage-like variable (activity), and a recovery-like variable (excitability), following similar approaches in the literature M. (2007). In this paper, we propose the following nonlinear oscillator model for each scalar component:

$$\dot{x}_{1i}(t) = \alpha_1 x_{1i}(t)(1 - x_{1i}{}^2(t)) + \alpha_2 x_{2i}(t) + \sum_{j=1}^{m} W_{ij} x_{1j}(t) \tag{1}$$

$$\dot{x}_{2i}(t) = -\alpha_3 x_{1i}(t)$$

where $x_{1i}(t)$ and $x_{2i}(t)$ represent the (observed) activity- and the (hidden) recovery-like variables of the $i$-th neural unit, respectively, and where the $W$ matrix represents the coupling strength between the observed variables, or neural units. The parameters $\alpha_k$ determine the bifurcation diagram of the system, allowing for a rich availability of possible dynamical states including oscillations and spike-like responses S. (2003); M. (2007).

Our ultimate goal is to learn the dynamical model (1), i.e. to estimate the parameters $\alpha_i, i = 1, ..., 5$ and $W_{ij}$, from calcium imaging data. We discretize the ODE model in equation (1), and formulate a joint inference problem for the state space $x$ and parameters $\alpha, W$ that is informed by noisy direct observations of some components; and constrained by the discretized dynamics.

**Parameter estimation.** We consider learning one component firs, and then extend the approach to multivariate model. Let $x^k \in \mathbb{R}^2$ denote a component of the van der Pol model given earlier in the equation (1), where $x^k = (x_1^k, x_2^k)$, i.e. a combination of both the observed and the hidden variables, and $k$ is the time index. The discretized dynamics governing the evolution of the single component $x = (x_1, x_2)$ can be written as $x^{k+1} = g(x^k, \alpha)$, where $g$ is a first-order Euler discretization of the nonlinear ODE (1). The same $\alpha$ inform the evolution of the entire time series $x = \begin{bmatrix} x_1^T & \cdots & x_N^T \end{bmatrix}^T$, which we write as $G(x, \alpha) = \eta^0$, with

$$G(x, \alpha) = \begin{bmatrix} x^1 \\ x^2 - g(x^1, \alpha) \\ \vdots \\ x^N - g(x^{N-1}, \alpha) \end{bmatrix}, \quad \eta^0 = \begin{bmatrix} x^0 \\ 0 \\ \vdots \\ 0 \end{bmatrix}. \tag{2}$$

Given noisy observations $z^k = H_k x^k + \omega_k$, we obtain consider ODE-constrained optimization problem

$$\min_{x,\alpha} \quad \frac{1}{2} \|z - Hx\|^2 \quad \text{s.t.} \quad G(x, \alpha) = \eta^0, \tag{3}$$

To solve this problem, we use a recent technique from PDE-constrained optimization, called the *penalty method* van Leeuwen & Herrmann (2015). The technique is a new take on the quadratic penalty method. Rewriting (6) with a quadratic penalty, we obtain the relaxed problem

$$\min_{x,\alpha} f_\lambda(x, \alpha) := \frac{1}{2} \|z - Hx\|^2 + \frac{\lambda}{2} \|G(x, \alpha) - \eta^0\|^2. \tag{4}$$

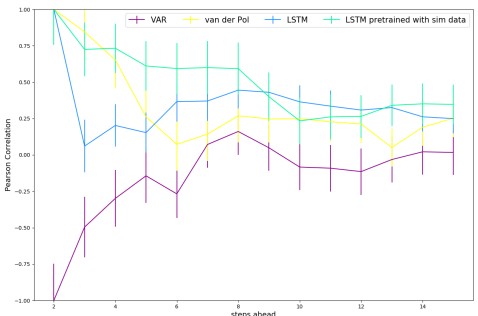

Figure 1: Predictive performance: correlation between the actual and predicted time series.

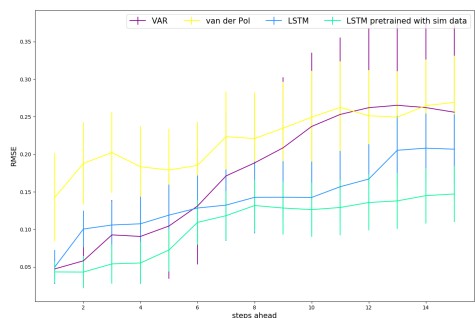

Figure 2: Predictive performance: root mean square error (RMSE) between the actual and predicted time series.

The key idea is to then use partial minimization with respect to $x$ at each iteration of $\alpha$ (see appendix for more detail).

Extending the model to $m$ components, and incorporating $W$ parameters besides $\alpha$, we obtain the coupled model shown previously in (1) (more details on the optimization are provided in the Appendix). Furthermore, we implemented a combination of the above optimization with stochastic search in order to improve parameter initialization before starting optimization; the combined approach, alternating between stochastic search and variable-projection yielded best results.

## 3 EXPERIMENTS

First, we evaluated multiple runs of van der Pol estimation procedure described above, and obtained high correlations between the actual data and the model's prediction, ranging from 0.76 to 0.83 for all six components, demonstrating that the model was able to capture well the underlying dynamics in the data.

We next compared our van der Pol model, LSTM and hybrid O-LSTM approach (which consists in simulating multiple times series from the van der Pol model estimated above and pretraining LSTM before training it on real data), as well as baseline Vector Auto-Regressive (VAR) model. LSTM: implementation details. Our LSTM networks, implemented in Keras, contained two layers, 128 units in each layer, followed by the fully-connected layer and linear activation; we used the mean squared error and the optimizer RMSProp; the drop-out rate was set to 0.8. We used a time window of 6 to predict the next time point.

O-LSTM was pretrained with 100 epochs using the above simulated data, and then trained with 50 epochs on the real training dataset; the number of epochs was selected so that the total number of samples used for training was the same for both simulated and real training data.

Figures 6 and 7 show the median correlation and the root-means square error, respectively, for several predictive methods: vector autoregressive (VAR) model (magenta), van der Pol model (yellow), LSTM (blue), and O-LSTM, i.e. LSTM pretrained on the data simulated using the above van der Pol model (green). Here we estimated parameters of the models on 100 consecutive points of training data, and then predicted the next 30 points (x-axis plots the index of the time points being predicted).

Note that the linear VAR model (magenta) performs poorly, unable to capture the nonlinear dynamics; both van der Pol (yellow) and LSTM (blue) have somewhat comparable performance, outperforming each other in shorter or in longer run; however the combination of both, O-LSTM model appears to be superior to both, achieving best accuracy for majority of future time points.

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

## 4 APPENDIX

## 5 ESTIMATING VAN DER POL PARAMETERS: KALMAN SMOOTHING-CONSTRAINED INFERENCE

We discretize the ODE model in equation (1), and formulate a joint inference problem for the state space $x$ and parameters $\alpha, W$ that is informed by noisy direct observations of some components; and constrained by the discretized dynamics.

### 5.1 INFERENCE FOR A SINGLE COMPONENT

Let $x^k \in \mathbb{R}^2$ denote a component of the van der Pol model given earlier in the equation (1), where $x^k = (x_1^k, x_2^k)$, i.e. a combination of both the observed and the hidden variables, and $k$ is the time index. The discretized dynamics governing the evolution of the single component $x = (x_1, x_2)$ can be written

$$x^{k+1} = g(x^k, \alpha),$$

where $g$ is a first-order Euler discretization of the nonlinear ODE (1). The same $\alpha$ inform the evolution of the entire time series $x = \begin{bmatrix} x_1^T & \cdots & x_N^T \end{bmatrix}^T$, which we write as $G(x, \alpha) = \eta^0$, with

$$G(x, \alpha) = \begin{bmatrix} x^1 \\ x^2 - g(x^1, \alpha) \\ \vdots \\ x^N - g(x^{N-1}, \alpha) \end{bmatrix}, \quad \eta^0 = \begin{bmatrix} x^0 \\ 0 \\ \vdots \\ 0 \end{bmatrix}. \tag{5}$$

Given noisy observations

$$z^k = H_k x^k + \omega_k,$$

we obtain consider ODE-constrained optimization problem

$$\min_{x, \alpha} \quad \frac{1}{2}\|z - Hx\|^2 \quad \text{s.t.} \quad G(x, \alpha) = \eta^0, \tag{6}$$

Problem (6) is challenging because (1) the ODE constraint function $G$ is nonlinear in $x$, and (2) because it is a joint optimization problem over $\alpha$. To solve this problem, we use a recent technique from PDE-constrained optimization, called the *penalty method* van Leeuwen & Herrmann (2015).

The technique is a new take on the quadratic penalty method. Rewriting (6) with a quadratic penalty, we obtain the relaxed problem

$$\min_{x,\alpha} f_\lambda(x,\alpha) := \frac{1}{2}\|z - Hx\|^2 + \frac{\lambda}{2}\|G(x,\alpha) - \eta^0\|^2. \tag{7}$$

The key idea is to then use partial minimization with respect to $x$ at each iteration of $\alpha$, thinking of the problem as follows:

$$\min_\alpha \widetilde{f}_\lambda(\alpha) := \min_x f_\lambda(x,\alpha).$$

The intuitive advantages of this method (find the best state estimate for each $\alpha$ regime) are borne out by theory. In particular, for a large class of models, the objective function $f_\lambda(\alpha)$ is well-behaved for large $\lambda$, unlike the joint objective $f_\lambda(x,\alpha)$ Aravkin et al. (2017)[1].

Evaluating $f_\lambda(\alpha)$ requires a minimization routine. We compute gradient and Hessian approximations

$$\nabla_x f_\lambda = H^T(Hx - z) + \lambda G_x^T(G(x,\alpha) - \eta^0)$$
$$\nabla_x^2 f_\lambda \approx H^T H + \lambda G_x^T G_x$$

where $G_x = \nabla_x G(x,\alpha)$. Evaluating $f_\lambda$ requires obtaining an (approximate) minimizer $\hat{x}$. With $\hat{x}$ in hand, $\nabla_\alpha f_\lambda$ can be computed using the formula

$$\nabla_\alpha f_\lambda(\alpha) \approx \lambda G_\alpha(\hat{x},\alpha)(G(\hat{x},\alpha) - \eta^0), \quad G_\alpha = \nabla_\alpha G(x,\alpha). \tag{8}$$

The accuracy of the inner solve in $x$ can be increased as the optimization over $\alpha$ proceeds. Constraints can also be placed on $\alpha$ if desired, to eliminate non-physical regimes or to incorporate prior information.

## 5.2   EXTENSION TO $m$ COMPONENTS

In addition to estimating the dynamic parameters $\alpha$, we are also interested in inferring the connectivity matrix $W$. Extending the model to $m$ components, we obtain the coupled model shown previously in (1). We discretize the entire dynamic model again using a first-order Euler scheme to get $g(x^k, \alpha, W)$, so that the full nonlinear process model $G$ now has form

$$G(x,\alpha,W) = \begin{bmatrix} x^1 \\ x^2 - g(x^1,\alpha,W) \\ \vdots \\ x^N - g(x^{N-1},\alpha,W) \end{bmatrix}, \tag{9}$$

with $x \in \mathbb{R}^{2mN}$.

The optimization approach for $m$ components is analogous to the single-component case with only $\alpha$ unknown. Specifically we consider

$$\min_{x,\alpha,W} f_\lambda(x,\alpha,W) := \frac{1}{2}\|z - Hx\|^2 + \frac{\lambda}{2}\|G(x,\alpha,W) - \eta^0\|^2,$$

and optimize this function by partially minimizing in $x$ and then working with the value function

$$\widetilde{f}_\lambda(\alpha,W) = \min_x f_\lambda(x,\alpha,W).$$

For the $m$-parameter case, we optimize over $x$ at each iteration using the Gauss-Newton method detailed in the previous section. The outer iteration is a fast projected gradient method for minimizing $\widetilde{f}_\lambda(\alpha,W)$ subject to simple bound constraints.

---

[1] Specifically, the Lipschitz constant of the gradient of $\widetilde{f}_\lambda(\cdot)$ stays bounded as $\lambda \uparrow \infty$, which is clearly false for $f_\lambda(\cdot,\cdot)$.

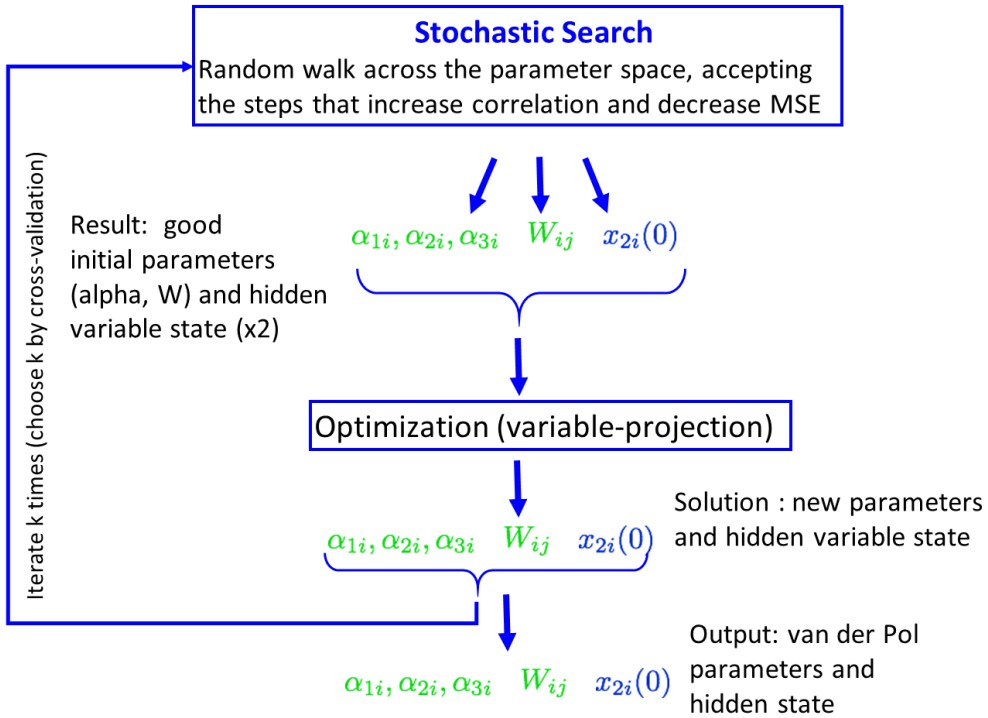

Figure 3: Van der Pol optimization procedure: variable-projection augmented with stochastic search.

# 6 LEARNING VAN DER POL MODEL: VARIABLE-PROJECTION + STOCHASTIC SEARCH

Optimizing van der Pol model can benefit considerably from a good initialization of its parameters, as we observed in multiple experiments. To improve initialization, we start with a random walk (stochastic search) in the parameter space, aiming at producing a reasonably good starting point for the optimization procedure; given a combination of parameters, we simulate time series using the corresponding van der Pol model, and measure the correlation and the mean-squared error between the simulated and the real (training) data, discarding the parameters whose performance metrics are under some threshold. Once a sufficiently high-performing model is found, we switch to the variable-projection (VP) method described above, initialized with the current parameters, which are now optimized even further; the whole process of alternating between random walk and VP optimization can be repeated several times, as shown in Figure 3. This combined procedure will be referred to in our subsequent section as simply *van der Pol optimization*.

**Stochastic search: implementation details.** We start with an initial guess for alphas (same value for all univariate oscillators), zero connectivity matrix $W$ and a random guess for the initial condition of the hidden variables, $x_2(0)$. At every stochastic search step, these parameters are updated in a certain stochastic way, described below, and the differential equations with the new parameters are integrated; if the resulting time-series solution improves the fit to the training data, the new parameters are accepted, otherwise they are dismissed. As a measure of the goodness of fit we use a linear combination of the Symmetric Mean Absolute Percentage Error and the Pearson correlation. In the first stage of our search, we only update $\alpha$s and $x_2(0)$, while keeping zero weight matrix $W$ (i.e., disconnected components). In each step, one of the components $x_i$ is chosen randomly, and its corresponding $\alpha$s and $x_2i(0)$ are changed using a Gaussian random walk. Also, once in a while, larger steps are taken (with greater than usual variance), in order to escape potential local minima. After the above initial stage, assuming a reasonably good parameters of individual oscillators are obtained, we start changing all parameters, including $W$. Unlike the rest of the parameters, all the

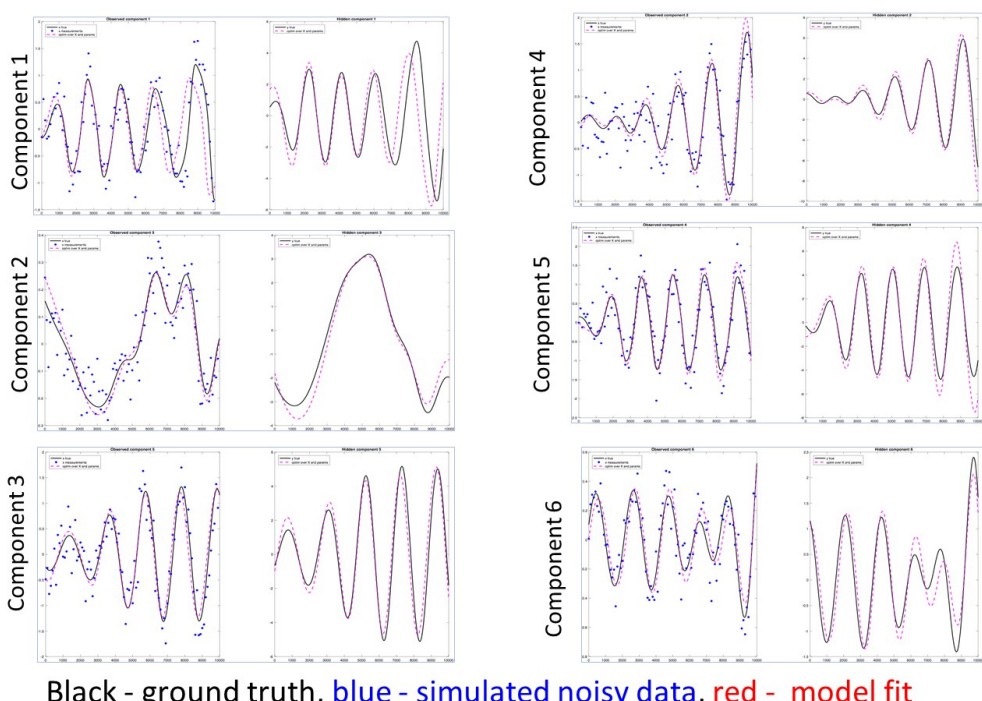

Black - ground truth, blue - simulated noisy data, red - model fit

Figure 4: Van der Pol model fit on simulated data.

components of $W$ change at every step, though normally using smaller (low-variance) random steps, in between less frequent higher-variance ones.

**VP and Stochastic Search combination: implementation details.** As described above, we estimate van der Pol parameters using a combination of stochastic search and variable-projection (VP) optimization. In our experiments, up to 200000 stochastic steps were used with a maximum of 50 outer iterations of the VP optimization every 1000 stochastic steps. The $W$ matrix remains zero for the first 15000 iterations. However the starting step for the optimization is random (uniformly distributed between 1 and 1000) in order to contribute for getting different resultant fittings, and associated parameters, after every run. Large stochastic steps are performed after every 30 small steps; the variance in $W$ steps increases by one order of magnitude (from 0.01 to 0.1); for $\alpha$ steps, the variances remains the same (0.1), but larger steps involve changing all components at once, rather than one at a time in smaller steps. After experimenting with various values of $\lambda$ parameter in the VP optimization, we chose $\lambda = 3e9$.

**Time series prediction.**
Once a van der Pol model is trained on a given time window, we can use it to predict the future time series, by integrating the model with the given parameters and the initial hidden state variable.

**Interpretability.**
Note that one of the advantages of the analytical van der Pol model is its interpretability, as it learns the interaction matrix $W$ among different spatial components, i.e. brain areas.

## 7    HYBRID APPROACH: OSCILLATOR-PRETRAINED LSTM (O-LSTM)

Clearly, an alternative to learning an analytical model, such as van der Pol, would be to use a general-purpose statistical model for time-series prediction, such as, for example, *Vector Autoregressive Model (VAR)* or recurrent neural network, in particular, LSTM networks. However, we found that the linear VAR model was not capturing well nonlinear dependencies in our data (as shown later in the

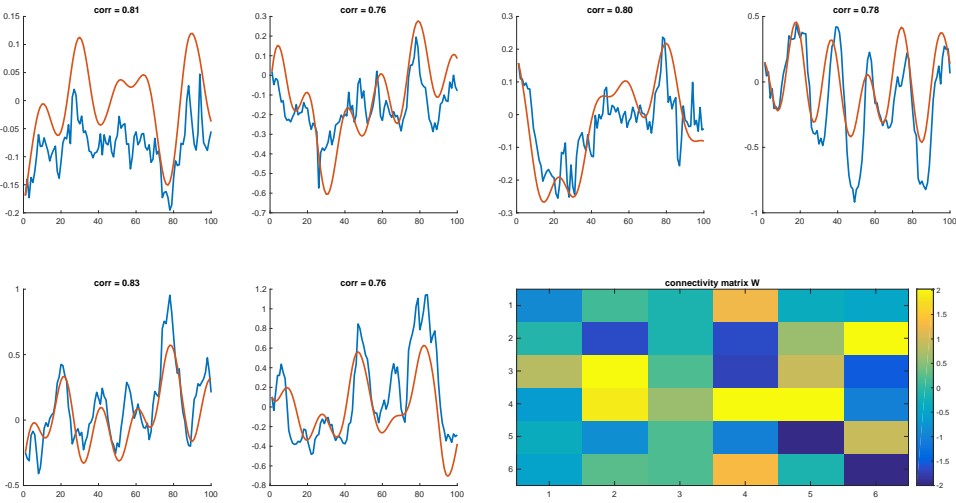

Figure 5: Van der Pol model fit on training data; correlations between the true and predicted time series for each of the six temporal SVD components. Bottom-right: an interaction matrix $W$.

empirical section), while LSTM networks were much more effective, while neither LSTM nor van der Pol model where dominating its rival in all settings; as we will see shortly, combining the two approaches led to the best performance. We will first provide a brief background on LSTM model.

### 7.1 BACKGROUND: LONG SHORT-TERM MEMORY (LSTM)

Recurrent neural networks take input in the shape of a sequence $x = (x_1, ..., x_T)$ and compute hidden vector sequence $h = (h_1, ..., h_T)$ and output vector $y = (y_1, ..., y_T)$ by iterating the following equations from $t = 1$ to $T$:

$$h_t = H(W_{xh}x_t + W_{hh}h_{t-1} + b_h) \tag{10}$$
$$y_t = W_{hy}h_t + b_y, \tag{11}$$

where the $W$ terms denote weight matrices, $b$ terms denote bias vectors, and $H$ is the hidden layer function.

Given the dynamic nature of neural responses and therefore EEG, using recurrent neural networks (RNN) is a reasonable choice for modeling brain activity's temporal evolution. Long Short-Term Memory (LSTM) (Hochreiter & Schmidhuber, 1997) is a RNN with improved memory. It uses memory cells with an internal memory and gated inputs/outputs which have shown to be more efficient in capturing long-term dependencies. The hidden layer function for LSTM is computed by the following set of equations:

$$i_t = \sigma(W_{xi}x_t + W_{hi}h_{t-1} + W_{ci}c_{t-1} + b_i) \tag{12}$$
$$f_t = \sigma(W_{xf}x_t + W_{hf}h_{t-1} + W_{cf}c_{t-1} + b_f) \tag{13}$$
$$c_t = f_t c_{t-1} + i_t \tanh(W_{xc}x_t + W_{hc}h_{t-1} + b_c) \tag{14}$$
$$o_t = \sigma(W_{xo}x_t + W_{ho}h_{t-1} + W_{co}c_t + b_o) \tag{15}$$
$$h_t = o_t \tanh(c_t), \tag{16}$$

where $\sigma$ is the logistic sigmoid function, and the components of the LSTM model, referred to as *input gate*, *forget gate*, *output gate* and *cell activation vectors* are denoted, respectively, as $i$, $f$, $o$, and $c$ (see (Hochreiter & Schmidhuber, 1997) for details).

LSTM: implementation details. Our LSTM networks, implemented in Keras, contained two layers, 128 units in each layer, followed by the fully-connected layer and linear activation; we used the mean squared error and the optimizer RMSProp; the drop-out rate was set to 0.8.

We used LSTM for multivariate time-series prediction, where each time point $t$ is represented by an $n$-dimensional vector (corresponding, in our case, to temporal components of the data at time $t$). We denote by $LSTM(k)$ the model which uses the previous $k$ time points to predict the $k + 1$-st time point. The prediction of the time step $k + 2$ is performed by shifting the window of length $k$ one step forward and using the prediction for the $k + 1$-st data point a new data point, iteratively. In our experiments, several values of $k$ were tried and $k = 6$ was selected.

## 7.2 LSTM WITH OSCILLATORY PRIOR (O-LSTM)

In general, given a relatively small amount of data for training LSTM, and a prior knowledge of some data properties, such as an oscillatory behavior, in our case, it would be best to find a way to inform LSTM about such a prior, i.e. to combine the generic statistical model such as LSTM with a domain-specific analytical model such as van der Pol.

We propose here a conceptually simple approach for introducing such prior into general-purpose LSTM: pretraining LSTM on a large amount of simulated data obtained from van der Pol, before "fine-tuning" LSTM on a relatively small amount of available real data, hoping that the data simulated from the analytical model will bias/regularize the LSTM model towards oscillating time series. More specifically, we propose the following procedure :

1. train $n$ van der Pol models (using the stochastic procedure in Figure 3) on the training data;
2. for each of those models, simulate $m$ noisy versions of time series (each of length $k$) obtained by integrating the model;
3. pre-train LSTM on the simulated data
4. continue LSTM training on real training data

**O-LSTM: implementation details.** Simulated data for pretraining LSTM contained 15 noisy versions of the 18 simulated time-series (160 time steps each), obtained by integrating the van der Pol equations with different sets of parameters estimated as described previosuly, by a combination of VP method and stochastic search.

LSTM was pretrained with 100 epochs using the above simulated data, and then trained with 50 epochs on the real training dataset; the number of epochs was selected so that the total number of samples used for training was the same for both simulated and real training data.

## 8 EXPERIMENTS

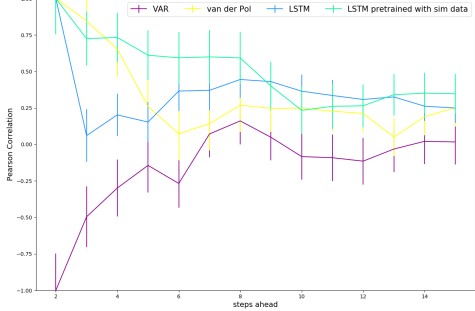

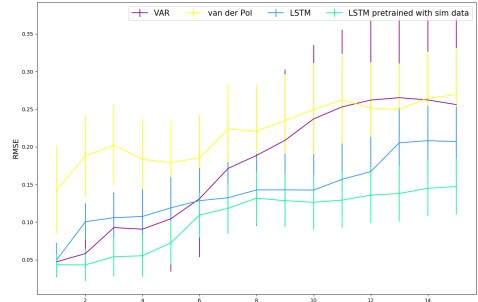

Figure 6: Predictive performance: correlation between the actual and predicted time series.

Figure 7: Predictive performance: root mean square error (RMSE) between the actual and predicted time series.

We now present our empirical results, including (1) simulated results; (2) the quality of van der Pol model fit on the training data, (3) predictive accuracy when forecasting time series using our van der

Pol model, LSTM and hybrid O-LSTM, as well as baseline Vector Auto-Regressive (VAR) model, and (4) a brief discussion of interpreting van der Pol interaction matrix.

## 8.1 SIMULATED DATA

First, we simulated time series using van der Pol models with a variety of parameter values, and provided our algorithm with the initial value of the hidden variable, just to assess the ability of our approach to recover hidden component which is unavailable in real data. Figure 4 presents the results for one of such experiments, where the recovered observed and hidden components match the actual ones very closely.

## 8.2 EVALUATING VAN DER POL MODEL FIT ON REAL DATA

Next, we evaluated multiple runs of van der Pol estimation procedure described above, combining stochastic search with VP optimization; Figure 5 shows the fit to the training data achieved by one of the best-performing model; we can see that the correlations between the actual data and the model's prediction are quite high, ranging from 0.76 to 0.83 for all six components, demonstrating that the model was able to capture well the underlying dynamics in the data.

In the right bottom corner of the figure, we plot the coupling matrix $W$. As mentioned before, this matrix represents the coupling strength between the different components, which, in turn, correspond to particular brain areas (spatial components shown before in Figure **??**). Thus, $W$ contains an interesting information about interactions (positive and negative) across different brain regions/subnetworks. For example, we oserve a strong positive interactions between the components 4 and 5, which correspond to the brain areas where an "flip-flop" oscillating behavior can be clearly observed (e.g., see the 2D version of the temporal data at https://www.youtube.com/watch?v=lppAwkek6DI).

## 8.3 PREDICTION ON TEST DATA

Figures 6 and 7 show the median correlation and the root-means square error, respectively, for several predictive methods: vector autoregressive (VAR) model (magenta), van der Pol model (yellow), LSTM (blue), and O-LSTM, i.e. LSTM pretrained on the data simulated using the above van der Pol model (green). Here we estimated parameters of the models on 100 consecutive points of training data, and then predicted the next 30 points (x-axis plots the index of the time points being predicted).

Note that the linear VAR model (magenta) performs poorly, unable to capture the nonlinear dynamics; both van der Pol (yellow) and LSTM (blue) have somehwat comparable perfromance, outperforming each other in shorter or in longer run; however the combination of both, O-LSTM model appears to be superior to both, achieving best accuracy for majority of future time points.

