# OpenReview forum: "Learning Oscillatory Brain Dynamics: van der Pol Meets LSTM"
_ICLR.cc/2018/Workshop — Reject_

### Official Review · AnonReviewer1 · 2018-03-05
**Interesting ideas but quite raw**

**Rating:** 5
**Confidence:** 4

**Review:**

The paper presents a method for modelling dynamic brain imaging data by first fitting a differential equation model to the data, using it to generate synthetic data for pre-training a LSTM model and finally training the LSTM on the very limited real data. The method is shown to yield more accurate predictions of future brain activity than different baselines.

I like the idea of using a differential equation model to generate pre-training data. I would, however, have expected a more meaningful prediction end-point, such as learning of connectivity or some other task that makes sense for the application. In terms of the method, it is unclear how the data were scaled (van der Pol model is not scale-invariant). The terminology in explaining the method is unclear, e.g. what is "variable projection"? Unnumbered eq. below Eq. (7) makes no sense as LHS is independent of alpha while RHS is not.

The text is very rough and contains a lot of mistakes and inaccuracies, e.g. many figure references and citations are bad. Please explain what the figure error bars denote.

---

### Official Review · AnonReviewer3 · 2018-03-09

**Rating:** 2
**Confidence:** 4

**Review:**

The authors consider linearly coupled van der Pol system with two state variables, three parameters and 6x6 couplings. They apply a standard Lagrangian relaxation of the ODE model against state observations.

The paper is confusing and lacks polish. There is a full 7 page article in the appendix, and the abstract is a confusing copy-paste from that. The work simply cannot be understood without the appendix.

The first part of the paper describes a stochastic parameter estimation as their model, while the experiments suddenly imply that the proposed model is actually LSTM neural network. The figure captions and method abbrevatiations don't match. The appendix reveals lots of material that is not even mentioned in the abstract.

Even for a workshop abstract, this abstract (the first three pages) is ineligible and not ready.

---

### Official Review · AnonReviewer2 · 2018-03-11

**Rating:** 7
**Confidence:** 3

**Review:**

The paper proposes a novel approach for learning a nonlinear dynamical model based on a variable-projection optimization approach that performs well on brain data.
The learned model is also successfully tested as a data generation mechanism for pre-training a LSTM.
I would advise the authors to re move the 'nonlinear Kalman smoothing' in the .pdf title, as the proposed method has nothing to do with Kalman smoothing.

---

### Decision · Program_Chairs · 2018-03-20
**ICLR 2018 Workshop Acceptance Decision**

**Decision:**

Reject

**Comment:**

Based on the reviews, this paper has not been accepted for presentation at the ICLR workshop. However, the conversation and updates can continue to appear here on OpenReview.